# High-Dynamic-Range Imaging for Cloud Segmentation

Soumyabrata Dev[1,3], Florian M. Savoy[2], Yee Hui Lee[1], and Stefan Winkler[2]

[1] School of Electrical and Electronic Engineering, Nanyang Technological University (NTU), Singapore 639798
[2] Advanced Digital Sciences Center (ADSC), University of Illinois at Urbana-Champaign, Singapore 138602
[3] ADAPT SFI Research Centre, Trinity College Dublin, Ireland

*Correspondence to:* Stefan Winkler (Stefan.Winkler@adsc.com.sg)

**Abstract.** Sky/cloud images obtained from ground-based sky-cameras are usually captured using a fish-eye lens with a wide field of view. However, the sky exhibits a large dynamic range in terms of luminance, more than a conventional camera can capture. It is thus difficult to capture the details of an entire scene with a regular camera in a single shot. In most cases, the circumsolar region is over-exposed, and the regions near the horizon are under-exposed. This renders cloud segmentation for such images difficult. In this paper, we propose HDRCloudSeg – an effective method for cloud segmentation using High-Dynamic-Range (HDR) imaging based on multi-exposure fusion. We describe the HDR image generation process and release a new database to the community for benchmarking. Our proposed approach is the first using HDR radiance maps for cloud segmentation and achieves very good results.

## 1 Introduction

Clouds are important for understanding weather phenomena, the earth's radiative balance, and climate change (IPCC, 2013; Stephens et al., 2012). Traditionally, manual observations are performed by *cloud experts* at WMO (World Meteorological Organization) stations around the world. Such manual methods are expensive and prone to human error. Weather instruments viz. ceilometers are useful in understanding the vertical profile of the cloud formation. However, they are point-measurement devices, and can provide cloud information along a particular slant path through the atmosphere. Satellite sensors are also extensively used in monitoring the earth's atmosphere. However, satellite images typically suffer from either low temporal or low spatial resolution.

Recently, ground-based cloud observations using high-resolution digital cameras have been gaining popularity. Whole Sky Imagers (WSIs) are ground-based cameras capturing images of the sky at regular intervals with a wide-angle (fish-eye) lens. They are able to gather high-resolution and localized information about the sky condition at frequent intervals. Due to their low cost and easy setup, their popularity among research groups working in several remote sensing applications is growing. One of the earliest sky cameras was developed by the Scripps Institute of Oceanography at the University of California San Diego (Kleissl et al., 2016). It was used to measure sky radiances at various wavelengths. Nowadays, commercial sky cameras are also available; however, they are expensive and offer little flexibility in their usage. Therefore, we have built our own custom-designed sky cameras with off-the-shelf components (Dev et al., 2014b). We use them extensively in our study of clouds in Singapore. The instantaneous data of cloud formations they provide is used for understanding weather phenomena, predicting

solar irradiance (Fua and Cheng, 2013; Dev et al., 2016), predicting the attenuation of communication links (Yuan et al., 2014) and contrail tracking (Schumann et al., 2013).

While there have been several studies analyzing clouds and their features from WSI images (Long et al., 2006; Souza-Echer et al., 2006; Li et al., 2011; Liu et al., 2015a; Dev et al., 2014a), most of them avoid the circumsolar region, because capturing the details in this area is a non-trivial task. The region around the sun has a luminous intensity several orders of magnitude higher than other parts of the scene. The ratio between the largest and the smallest luminance value of a scene is referred to as its Dynamic Range (DR). On a typical sunny and clear day, it is difficult to capture the entire luminance range of the sky scene using a low-dynamic-range (LDR) image. Yankee Environmental Systems sells a well-known commercial sky camera model (TSI-880) (Long et al., 2001). It is a fully automated sky imager, used in continuous monitoring of the clouds. Its on-board processor computes the cloud coverage and sunshine duration, and stores these results for the user for further processing. However, such imagers only capture low-dynamic-range (8-bit) images.

One of the earliest attempts to capture more of the dynamic range of the sky was done by Stumpfel et al. (2004). They presented a framework in which a set of exposure settings along with neutral density filters are used to generate an HDR composite map. Kenny et al. (2006) used a digital camera to estimate the whole-sky luminance distribution for different sky conditions. Attempts to provide a full spherical HDR view of the sky/cloud condition were done by mounting hemispherical sky cameras on the top and bottom of airships (Okura et al., 2012). Gryaditskya et al. (2014) used HDR captures of the sky to recover absolute luminance values from images. To the best of our knowledge, there is no prior work that uses the capability of HDR imaging for better segmentation of sky/cloud images.

Several techniques for cloud segmentation exist, but they are designed for conventional LDR images. Long et al. (2006) developed a method based on fixed thresholding. As clouds have a non-rigid structure, traditional segmentation algorithms based on shape priors are not applicable. Color is generally used as a discriminating feature in cloud segmentation (Li et al., 2011; Souza-Echer et al., 2006; Mantelli-Neto et al., 2010). Li et al. (2011) use a hybrid thresholding approach that employs both fixed and adaptive thresholds, depending on the bimodality of the input image. Long et al. (2006) model atmospheric scattering by calculating the ratio of red and blue color channels. Souza-Echer et al. (2006) define appropriate thresholds in the saturation channel of the intensity-hue-saturation (IHS) color model. Mantelli-Neto et al. (2010) uses a multi-dimensional Euclidean distance to determine the locus of sky and cloud pixels.

The motivation of this paper is to propose high-dynamic-range imaging for cloud segmentation using ground-based sky cameras. We detail the process of image capture and storage in our sky camera system. We show that using HDR cloud imaging significantly reduces the amount of saturated pixels in an image, and is therefore an efficient manner to capture the circumsolar region. Furthermore, HDR imaging generally provides better segmentation results, as compared to LDR images, regardless of the segmentation method used. In this paper, we show how to improve segmentation results by capturing a larger dynamic range of the sky using High-Dynamic-Range Imaging (HDRI) techniques. We then introduce HDRCloudSeg, a graph-cut based segmentation algorithm that uses the HDR radiance map for accurate segmentation of sky/cloud images.

Figure 1 summarizes our proposed approach.

The main novel contributions of the present manuscript include:

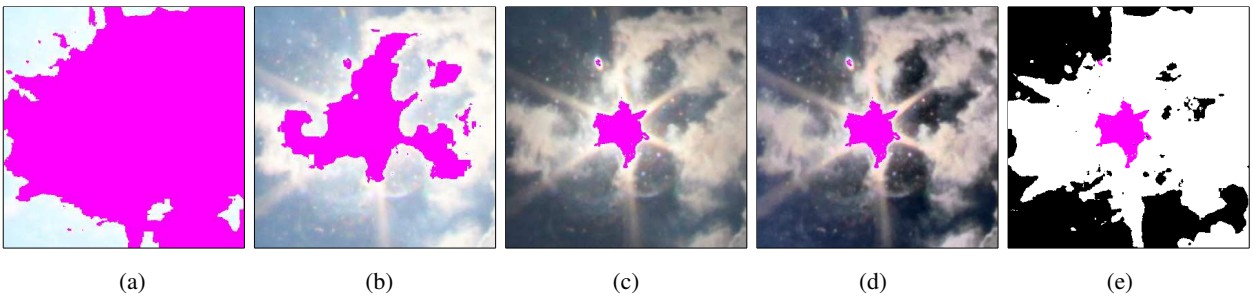

| (a) | (b) | (c) | (d) | (e) |

**Figure 1.** Proposed HDRCloudSeg cloud segmentation approach. (a) High- (b) Medium- (c) Low-exposure Low-Dynamic-Range (LDR) images; (d) High-Dynamic-Range (HDR) image (tonemapped for visualization purposes); (e) Binary output of HDRCloudSeg. Saturated pixels in all images are shown in pink. The number of saturated pixels is significantly reduced in the HDR image, without compromising on the fine cloud details.

– Introducing an HDR capture process for better cloud imaging as compared to conventional LDR images. This includes methods and procedures to capture a larger part of the dynamic range of a sky scene and archiving the obtained images efficiently;

– Proposing a graph-cut based segmentation algorithm that offers better performance as compared to the state-of-the-art approaches;

– Releasing a database comprising high dynamic captures of the sky scene, along with their manually annotated ground-truth segmentation maps.

The rest of this paper is structured as follows. We discuss the process of HDRI generation in Section 2. We introduce HDRCloudSeg in Section 3 and evaluate it in Section 4. Section 5 concludes the paper.

## 2 HDR Image Generation

WAHRSIS, our custom-designed sky imagers, are composed of a Digital Single-Lens Reflex (DSLR) camera with a fish-eye lens. The camera is controlled by a single-board computer housed inside a box with a transparent dome. Our first model (Dev et al., 2014b) used a moving sphere mounted on two motorized arms to block the direct light of the sun. We removed this from our latest models in favor of HDR imaging (Dev et al., 2015b). In this way, we avoid potential mechanical problems as well as occlusions in the resulting images.

Typical cameras can only capture an 8-bit low dynamic range (LDR) image. Therefore, we use the exposure bracketing option of our cameras to capture three pictures at $0$, $-2$ and $-4$ exposure value (EV) in quick succession. The aperture remains fixed across all captured images, while the shutter speed automatically adapts to match the appropriate exposure. Figure 2 shows an example of the captured images at varying exposure levels. An LDR image taken at low exposure (cf. Fig.2(c)) has few saturated pixels in the circumsolar region, but is underexposed in the regions further from the sun. On the other hand, a

high- or medium-exposure LDR image (cf. Fig.2(a) and (b)) can capture the near-horizon regions well, but the circumsolar area is overexposed and saturated.

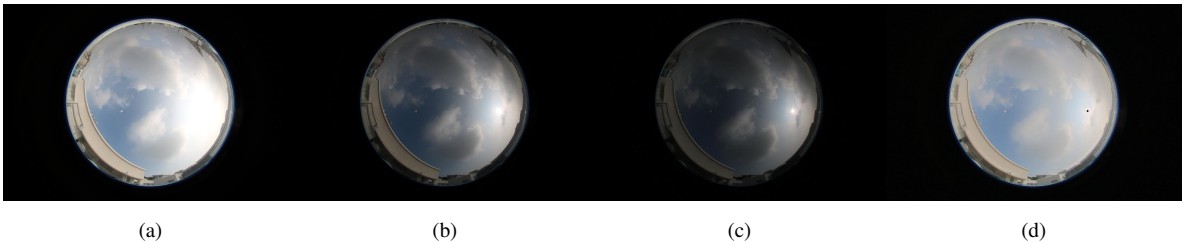

(a)  (b)  (c)  (d)

**Figure 2.** Example of three images captured by the camera at varying exposure levels and shutter speeds. (a) 0 EV, 1/400 sec. (b) -2 EV, 1/1600 sec. (c) -4 EV, 1/4000 sec. (d) HDR image computed by fusing the pictures (a-c) (tonemapped for visualization purposes).

The different LDR images can then be fused together into a single, high-dynamic-range (HDR) radiance map. We use Debevec and Malik (1997)'s algorithm to recover the HDR radiance map from LDR images. While the radiance map can be analyzed computationally, it cannot be viewed directly on a conventional (LDR) display. Therefore, for the purpose of visualization, we tone-map the HDR radiance map into a conventional 8-bit LDR image using contrast-limited adaptive histogram equalization (CLAHE) (Zuiderveld, 1994). It involves a logarithmic transformation of the higher DR radiance map to a lower 8-bit display. This produces a perceptually recognizable image, because the response of the human eye to light is also logarithmic. This operation compresses the dynamic range by preserving the details in the image. Figure 2(d) shows an example of a tone-mapped image.

We also perform the HDR fusion and tone-mapping on a server that collects the images from all our WSIs. Since we have three imagers capturing multiple exposures every 2 minutes, we need algorithms which are computationally efficient. Therefore, we use the GPU implementation as described by Akyüz (2015). It relies on the OpenGL API to perform the entire HDR pipeline on the GPU. It consists of HDR fusion, which is detailed in Akyüz (2015), followed by tone-mapping, which is based on the photographic tone reproduction operator of Reinhard et al. (2002). Generating an 18-Megapixel HDR image and its tone-mapped version takes less than 7 seconds, compared to several minutes with standard CPU algorithms.

For storage efficiency, we compress our HDR images using the *JPEG-XT* format (Richter, 2013).[1] *JPEG-XT* is an extension of *JPEG* currently being developed for HDR images. An important advantage of *JPEG-XT* format is that it is backward compatible to the popular *JPEG* compression technique. It consists of a base layer that is compatible with the legacy systems and an extension layer that provides the full dynamic range. Using JPEG-XT at a quality level of 90%, we obtain a file size of about 8 MB. This represents a significant improvement compared to the common *RGBE* format, which would require 50MB per image by storing every pixel with one byte per color channel and one byte as a shared exponent.

---

[1] We use the reference software source code available at `https://jpeg.org/jpegxt/software.html`

## 3 Cloud Segmentation Using HDRI

We propose a graph-based segmentation algorithm called HDRCloudSeg that formulates the sky/cloud segmentation task as a graph-partitioning problem. Graph-cut for cloud segmentation was proposed earlier by Liu et al. (2015b). However, Liu et al. used conventional LDR images in their segmentation framework and generated seeds using Otsu threshold (Otsu, 1979). Furthermore, they did not consider the circumsolar regions in their evaluation.

### 3.1 Notation

Suppose that the low-, medium- and high-exposure LDR sky/cloud images are represented by $\mathbf{X}_i^L$, $\mathbf{X}_i^M$ and $\mathbf{X}_i^H$ respectively, $i = 1, \ldots, N$, where $N$ is the total number of HDR sets in the dataset. Without any loss of generality, $\mathbf{X}_i$ denotes a low-, medium-, or high-exposure LDR image in the subsequent sections. Each of these LDR images are *RGB* color images, $\mathbf{X}_i \in \mathbb{R}^{a \times b \times 3}$, with dimension $a \times b$ for each *R*, *G* and *B* channel. We generate the HDR radiance map $\mathcal{H}_i \in \mathbb{R}^{a \times b \times 3}$ from the set of three LDR images $\mathbf{X}_i^L$, $\mathbf{X}_i^M$ and $\mathbf{X}_i^H$ as described in Section 2.

Let $p_{mn}$ be a sample pixel in the image $\mathbf{X}_i$, where $m = 1, \ldots, a$ and $n = 1, \ldots, b$. We aim to assign labels to each of the pixels $p_{mn}$, as either *cloud* or *sky*. We denote the label as $L_p$, where $L_p = 1$ or 0 if $p_{mn}$ is a *cloud* or *sky* pixel, respectively. We model this task as a graph-based discrete labeling problem, wherein we represent $\mathbf{X}_i$ as a graph, comprising a set of nodes and edges.

### 3.2 Generating Seeds

Graph-cut based segmentation algorithms (Boykov et al., 2001; Kolmogorov and Zabin, 2004; Boykov and Kolmogorov, 2004; Bagon, 2006) require the user to initially label a few pixels as 'foreground' and 'background'. We refer to these prior labeled pixels as *seeds*. The process of generating seeds is generally done manually before partitioning the graph into two sub-graphs. In HDRCloudSeg, we automatically generate these initial seeds by assigning a few pixels in the sky/cloud image with *sky* and *cloud* labels.

Most sky/cloud segmentation algorithms (Li et al., 2011; Long et al., 2006; Souza-Echer et al., 2006; Mantelli-Neto et al., 2010) use color as the discriminating feature to distinguish cloud from sky. In our previous work, we have performed a systematic analysis of existing color spaces and components for conventional LDR images and observed that color channels such as $(B - R)/(B + R)$ ($B$ and $R$ indicating the blue and red channels of the image) are among the most discriminative for cloud segmentation. We discuss further details on discriminatory color channels of HDR luminance maps in Section 4.2.1. Furthermore, we have proposed a probabilistic approach for sky/cloud image segmentation, instead of the conventional binary approach. In this probabilistic approach, each pixel is assigned a *soft* membership to belong to the *cloud* category, instead of a *hard* membership (Dev et al., 2014a).

To illustrate these concepts, consider the example shown in Fig. 3. Figure 3(a) shows a sample LDR sky/cloud image $\mathbf{X}_i$. We extract the $(B - R)/(B + R)$ color channel from this LDR image as shown in Fig. 3(b). A fuzzy C-means clustering on this extracted color channel yields the probabilistic output image, shown in Fig. 3(c). It denotes the confidence level of each pixel

to belong to the cloud category. We assume that for a given pixel, the sum of membership values for sky and cloud category is unity.

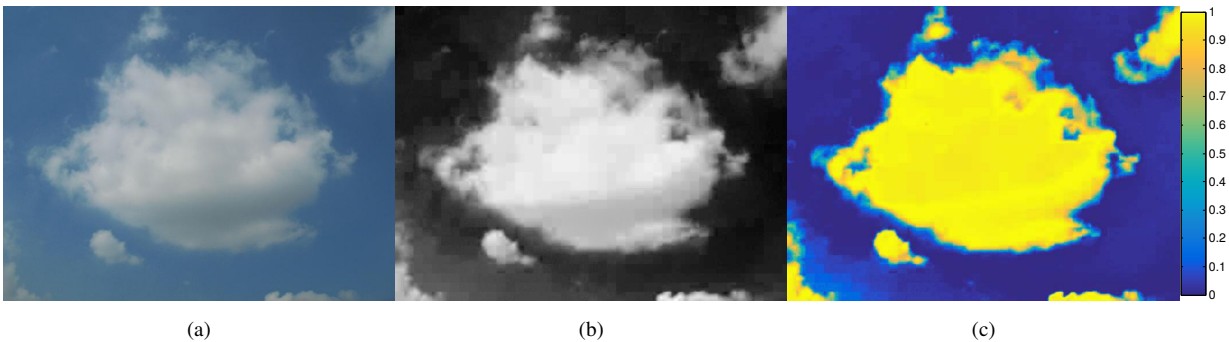

(a)                                         (b)                                         (c)

**Figure 3.** Illustrative example to demonstrate the probability of a pixel belonging to the 'cloud' category. (a) Sample sky/cloud image; (b) Normalized $(B-R)/(B+R)$ color channel image; (c) Probabilistic output image using the method from (Dev et al., 2014a).

In HDRCloudSeg, we apply fuzzy C-means clustering to the most discriminatory color channel of the HDR radiance map $\mathcal{H}_i$ to estimate the probability of a pixel belonging to the cloud category. We denote the ratio channel as $\mathbf{Y}_i$. The membership values obtained by clustering provide us a mechanism for assigning the seeds with a degree of confidence. We assign these initial seeds for HDRCloudSeg as follows: pixels having membership $>\alpha$ or $<(1-\alpha)$ are labeled as cloud and sky respectively. The value of $\alpha$ is a constant and is set experimentally (more on this below).

### 3.3   Partitioning the HDR Graph

HDRCloudSeg employs a graph-based image segmentation approach, wherein we represent the ratio-image $\mathbf{Y}_i \in \mathrm{I\!R}^{a \times b}$ as a set of nodes and edges. Each edge of the graph is given a corresponding weight that measures the dissimilarity between two pixels. Such methods are based on *pixel adjacency graphs*, where each vertex is a pixel and the edges between them are defined by adjacency relations. We follow the work of Boykov and Jolly (2001) and try to minimize the segmentation score $E$:

$$E = \sum_{p \in \mathcal{P}} R_p(A_p) + \mu \sum_{(p,q) \in \mathcal{N}; A_p \neq A_q} B_{p,q}, \tag{1}$$

where $R_p(A_p)$ denotes the data cost for an individual pixel $p$, $B_{p,q}$ denotes the interaction cost between two neighboring pixels $p$ and $q$ in a small neighborhood $\mathcal{N}$, and $\mu$ is a weight.

The complete proposed HDR segmentation algorithm is summarized in Algorithm 1.

We illustrate the complete HDRCloudSeg segmentation framework in Fig. 4. Figure 4(a) represents the three sample LDR images $\mathbf{X}_i^L$, $\mathbf{X}_i^M$ and $\mathbf{X}_i^H$ respectively captured at varying EV settings. We generate the corresponding HDR radiance map $\mathcal{H}_i$ from these LDR images. A tone-mapped version of $\mathcal{H}_i$ is shown in Fig. 4(b), for visualization purposes. We extract the $(B-R)/(B+R)$ ratio channel from $\mathcal{H}_i$, and generate the initial cloud and sky seeds marked in *green* and *red* color respectively, as shown in Fig. 4(c). The binary output image after graph cut is shown in Fig. 4(d).

**Algorithm 1** HDRCloudSeg

---

**Require:** LDR sky/cloud images with varying EVs.

1: Create HDR radiance map $\mathcal{H}_i$ from bracketed set of LDR images;

2: Extract $(B-R)/(B+R)$ ratio channel $\mathbf{Y}_i$ from HDR radiance map $\mathcal{H}_i$;

3: Perform fuzzy C-means clustering on the extracted ratio channel $\mathbf{Y}_i$ from HDR radiance map $\mathcal{H}_i$, to generate the probabilistic map;

4: Generate initial seeds from the computed probabilistic map for image segmentation as described in Section 3.2;

5: Partition the ratio channel $\mathbf{Y}_i$ into two subgraphs using the generated seeds;

6: **return** Binary segmented image.

---

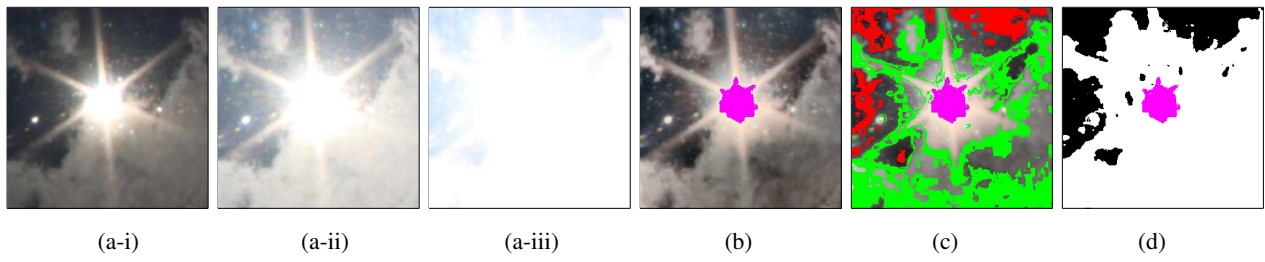

| (a-i) | (a-ii) | (a-iii) | (b) | (c) | (d) |

**Figure 4.** Proposed HDRCloudSeg framework for sky/cloud segmentation. (a) Exposure bracketed Low Dynamic Range (LDR) images; (b) High Dynamic Range (HDR) image tone-mapped for visualization purpose; (c) Image with initial seeds, where *cloud* seeds and *sky* seeds are represented in *green* and *red* respectively; (d) Binary sky/cloud segmentation result. The saturated pixels in (b-d) are masked in pink.

## 4  Experimental Evaluation

### 4.1  HDR Sky/Cloud Image Database

Currently there are no available HDR image datasets for sky/cloud images. Therefore, we propose the first HDR sky/cloud dataset to the research community. We refer to this dataset as Singapore HDR Whole Sky Imaging SEGmentation dataset (SHWIMSEG). The SHWIMSEG dataset consists of $52$ sets of HDR captures, comprising a total of $156$ LDR images. Each HDR capture is based on three LDR images (low-, medium- and high-exposure) which were captured in Automatic Exposure Bracketing (AEB) mode of our camera. These high-quality HDR images are captured with our sky imagers, located on the rooftops of a building at Nanyang Technological University Singapore. We crop a square region of $500 \times 500$ pixels from each images, with the sun location as the geometrical center of the cropped image. The corresponding ground truth images for these crops were manually generated in consultation with experts from the Meteorological Service Singapore (MSS).

We release this entire HDR image dataset on sky/cloud image segmentation.[2] The SHWIMSEG dataset comprises the following: (a) original full-size LDR fish-eye images captured in three exposure settings, (b) corresponding $500 \times 500$ cropped images of the circumsolar region, in all the exposure settings, (c) HDR radiance maps of the crops generated via exposure fusion, (d) tonemapped images (used during visualization purposes), and (e) manually annotated binary ground-truth maps

---

[2] The SHWIMSEG database is available online at `http://vintage.winklerbros.net/shwimseg.html`.

for the cropped images. We use this newly created dataset to perform a detailed evaluation of several cloud segmentation algorithms below.

## 4.2 Results

HDR imaging is an effective technique for cloud observation, as it helps us better image the circumsolar region with reduced overexposure. We illustrate the advantage of HDR imaging in reducing the saturation by calculating the number of saturated pixels in the low-, medium- and high-exposure LDR images. We also calculate the number of saturated pixels (if any) in the radians maps in the dataset.[3] Using our HDR techniques, we observe that the radiance maps have $24$ times fewer saturated pixels, as compared to the high-exposure LDR images, and $4$ times fewer saturated pixels with respect to medium-LDR images. A reduced amount of saturated pixels is an important factor for cloud analysis, especially around the circumsolar region.

In addition to containing fewer saturated pixels, HDR imaging also helps in improved cloud segmentation performance, regardless of the techniques used, as we will demonstrate in the following. Cloud segmentation is essentially a binary classification problem, wherein we classify each pixel as either sky or cloud. We measure the classification performance of the different cloud segmentation methods using Precision, Recall, F-score and Error values, which are defined as:

$$\text{Precision} = TP/(TP + FP),$$

$$\text{Recall} = TP/(TP + FN),$$

$$\text{F-score} = \frac{2 \times \text{Precision} \times \text{Recall}}{\text{Precision} + \text{Recall}},$$

$$\text{Error} = (FP + FN)/(TP + TN + FP + FN),$$

where *TP*, *TN*, *FP*, and *FN* denote the true positive, true negative, false positive and false negative samples in a binary image.

For evaluation purposes, we benchmark HDRCloudSeg with existing cloud segmentation approaches by Li et al. (2011), Long et al. (2006), Souza-Echer et al. (2006), and Mantelli-Neto et al. (2010). All these methods, which were briefly described in Section 1, are designed for conventional LDR images.

### 4.2.1 Color Channel Selection

Like in our previous work (Dev et al., 2014a), we study various color channels and models that are conducive for sky/cloud image segmentation. We consider $16$ color models and components that are generally used in sky/cloud image segmentation. Table 1 lists the various color channels: it contains the color models $RGB$, $HSV$, $YIQ$, $L^*a^*b^*$, various red and blue ratio channels, and chroma (Dev et al., 2017).

We designed our proposed HDRCloudSeg segmentation algorithm to work on the HDR radiance maps. We extract the $16$ color channels (as indicated in Table 1) from the HDR radiance map and perform fuzzy C-means clustering on the extracted

---

[3] In an LDR image, we consider a particular pixel as saturated if its value for all of the red-, green- and blue- color channels is greater than 250. In the case of the HDR radiance map, a pixel is considered saturated if it is saturated in all its corresponding low-, medium- and high-exposure LDR images.

| $c_1$ | R | $c_4$ | H | $c_7$ | Y | $c_{10}$ | $L^*$ | $c_{13}$ | $R/B$ | $c_{16}$ | $C$ |
|---|---|---|---|---|---|---|---|---|---|---|---|
| $c_2$ | G | $c_5$ | S | $c_8$ | I | $c_{11}$ | $a^*$ | $c_{14}$ | $R-B$ | | |
| $c_3$ | B | $c_6$ | V | $c_9$ | Q | $c_{12}$ | $b^*$ | $c_{15}$ | $\frac{B-R}{B+R}$ | | |

**Table 1.** Color spaces and components used in our analysis. We intend to find the best color channel for HDR imaging.

color channel. We assign the initial seeds for *cloud* and *sky* pixels in the fuzzy-clustered image (more details on the seeding level in the subsequent Section 4.2.2).

Figure 5 shows the segmentation error for all the 16 color channels. We observe that the saturation color channel ($c_5$) and the ratio color channel ($c_{15}$) has better performance, as compared to other color channels. Therefore, we choose the $c_{15} = (B-R)/(B+R)$ channel as the optimum color channel for HDR segmentation and in subsequent experiments. The $c_{15}$ color channel performs the best, owing to a physical phenomenon called Rayleigh scattering: small particles in the atmosphere scatter light at varying degree. The blue component (lowest wavelength) gets scattered the most, which renders a bluish color to the sky. The $c_{15}$ color channel is the normalized ratio of *red* and *blue* color channels; and is the most *discriminatory* feature for cloud detection.

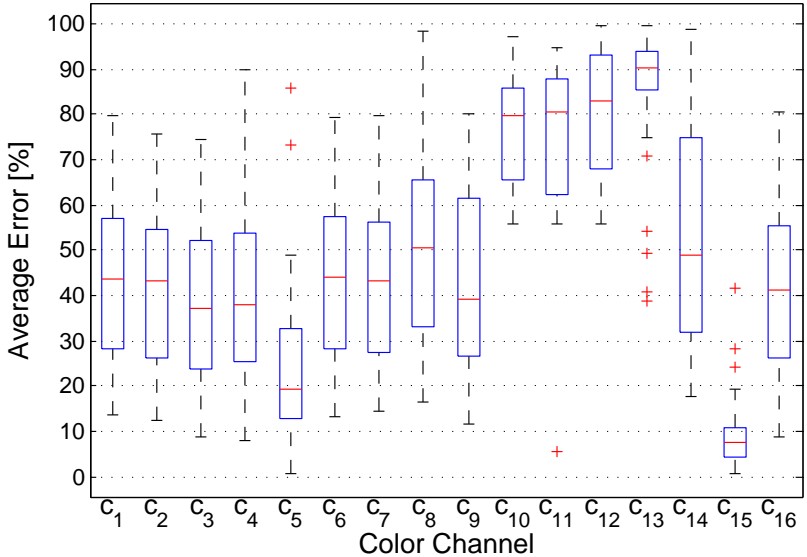

**Figure 5.** Distribution of the average segmentation error for different color channels of the HDR radiance map, across all the images of the HDR dataset. Within each box, the red line indicates the median, bottom and top edges indicate the 25th and 75th percentiles, respectively.

### 4.2.2   Seeding Level Sensitivity

As described in Section 3.2, the value of $\alpha$ determines the amount of initial seeds in HDRCloudSeg. We set a high value of $\alpha$, because it corresponds to higher confidence in assigning the correct labels. A higher confidence ensures low error and high

accuracy. We show the Receiver Operating Characteristics (ROC) Curve of our proposed algorithm, for varying values of the seeding parameter. The ROC curve in Figure 6(a) plots the False Positive Rate (FPR) (= $\frac{FP}{FP+TN}$), against the True Positive Rate (TPR) (= $\frac{TP}{TP+FN}$). We generate these points for varying values of $\alpha$ in the interval $[0,1]$. This curve helps the user to select the best value of the seeding parameter. In order to illustrate its effect on the segmentation performance, we also evaluate
the dependency of the seeding parameter on the various evaluation metrics.

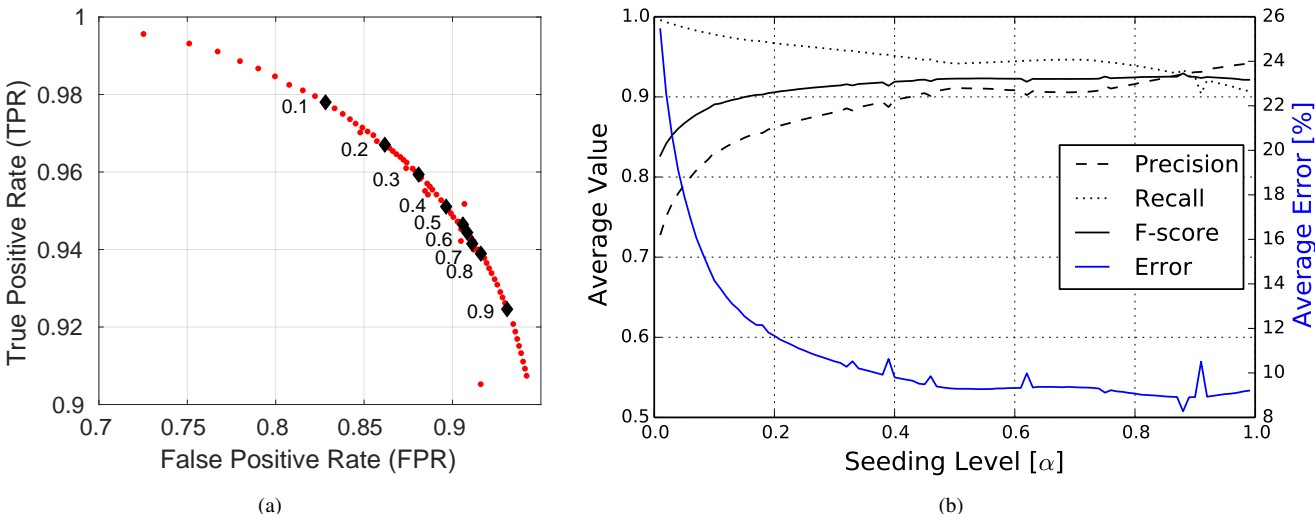

(a)                                                (b)

**Figure 6.** (a) Impact of the seeding parameter on Receiver Operating Characteristics (ROC) curve; the black diamonds indicate the intermediate ROC points, for seeding parameter in intervals of 0.1 in the range [0,1]. (b) Impact of the seeding parameter on Average Precision, Recall, F-score value and Error.

Figure 6 shows the impact of the seeding parameter $\alpha$ on the average performance of our segmentation framework. We report the average error percentage, precision, recall and F-score across all the images of the dataset. We observe from Fig. 6(b) that the average error gradually decreases with increasing value of the seeding parameter, $\alpha$. This makes sense as higher value of $\alpha$ indicate higher confidence for accurate detection of labels. Similar observations apply to average precision, recall and F-score.
The sensitivity of the seeding parameter can also help us in understanding different types of clouds around the sun aureole. In this paper, we deal only with the detection of thick clouds in the circumsolar region. Primarily this is because classifying the type of clouds into *thick* or *thin* clouds is an extremely difficult task in the area around the sun. Most of circumsolar region pixels are close to saturation. However, the seeding parameter $\alpha$ has a control on the type of detected clouds. For example, we can detect thin clouds by tuning the seeding parameter to favor higher cloud detection (i.e. more tendency to classify a pixel as
cloud), which comes at the expense of reduced precision, however.
      From Fig. 6(b), we observe that there is a dip in the Error at $\alpha = 0.88$. Moreover, at this value, there is a good trade-off between precision and recall values – both are high. Therefore, we set the value of the seeding parameter $\alpha$ to $0.88$ in the subsequent experiments. We also observe that there are a few deviation points along the curve. The minor peaks and troughs in

Fig. 6 are due to the sensitivity of the considered seeding level. This occasionally causes errors in the seeding accuracy, which subsequently impacts the final evaluation metric of cloud detection.

### 4.2.3 Segmentation Performance

Since existing cloud segmentation algorithms are designed for conventional LDR images, we evaluate them on the mid-exposure LDR images as well as the tonemapped HDR images. Our proposed HDRCloudSeg algorithm is the only one designed to make use of the full HDR radiance maps. However, for the sake of comparison, we also evaluate it for mid-exposure LDR and tonemapped images. The detailed evaluation results of HDRCloudSeg, along with the other cloud segmentation methods, are shown in Table 2.

| Methods | Image Type | Precision | Recall | F-score | Error [%] |
|---|---|---|---|---|---|
| Long et al. (2006) | LDR | 0.65 | **1.00** | 0.77 | 35.4 |
| | Tonemapped | 0.71 | 0.99 | 0.82 | 27.7 |
| Souza-Echer et al. (2006) | LDR | 0.68 | 0.99 | 0.79 | 31.0 |
| | Tonemapped | 0.85 | 0.97 | 0.89 | 14.4 |
| Mantelli-Neto et al. (2010) | LDR | 0.65 | **1.00** | 0.77 | 35.2 |
| | Tonemapped | 0.68 | 0.99 | 0.79 | 32.1 |
| Li et al. (2011) | LDR | 0.90 | 0.85 | 0.86 | 16.2 |
| | Tonemapped | 0.77 | 0.99 | 0.85 | 21.8 |
| Proposed | LDR | 0.86 | 0.94 | 0.89 | 12.8 |
| | Tonemapped | 0.83 | 0.97 | 0.89 | 15.0 |
| | HDR | **0.93** | 0.93 | **0.93** | **8.91** |

**Table 2.** Benchmarking results. The average scores across all the images are reported for the various methods. The best performance according to each criterion is indicated in bold.

From Table 2, we observe that HDR imaging improves the cloud segmentation performance irrespective of the method used. We observe that most of the benchmark algorithms (except for Li et al.) have a better performance with tonemapped HDR images as compared to the mid-exposure LDR image. This is because a tonemapped version exhibits fewer saturated pixels and clearer contrast between sky and cloud, as compared to the corresponding LDR image. However, our proposed method HDRCloudSeg using the entire HDR radiance map achieves the lowest error of $8.91\%$ across all the methods.

Most of the other algorithms are biased towards a higher recall value (tendency to over-estimate cloud cover) with lower precision. These existing algorithms are based on a set of thresholds, either fixed or adaptive, and are therefore more prone to high error percentage. However, HDRCloudSeg uses the entire dynamic range of sky/cloud scenes to make a more informed decision in classifying a pixel as cloud or sky.

## 5 Conclusions & Future Work

In this paper, we have proposed a novel approach to solve cloud segmentation in images captured by WSIs, using High Dynamic Range Imaging (HDRI) techniques. This greatly reduces post-processing steps (image inpainting and de-saturation), compared to sky camera designs based on a sun-blocker. These HDR images capture significantly more detail than traditional Low Dynamic Range (LDR) images, especially in the circumsolar region and near the horizon. We have shown that this method outperforms others. Our proposed methodology is reliable, efficient, and easy to be deploy.

In our future work, we plan to investigate how High Dynamic Range imaging improves the accuracy of other tasks, such as cloud classification (Dev et al., 2015a) or cloud height estimation (Savoy et al., 2015). In order to improve benchmarking, we will also work on expanding the sky/cloud HDR dataset introduced here with more images and distinctions between cloud types.

## 6 Database and Code

In the spirit of reproducible research (Vandewalle et al., 2009), we release the entire HDR sky/cloud image dataset as well as the source code of all simulations in this paper. The SHWIMSEG database is available at `http://vintage.winklerbros.net/shwimseg.html`; the Matlab code of HDRCloudSeg is available at `https://github.com/Soumyabrata/HDR-cloud-segmentation`.

*Acknowledgements.* This research is funded by the Defence Science and Technology Agency (DSTA), Singapore.

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
