# Peer review of "High-Dynamic-Range Imaging for Cloud Segmentation"

_Atmospheric Measurement Techniques, 2017_

## Referee Comment (RC1) · Anonymous Referee #1 · 22 Dec 2017

In the manuscript "High-Dynamic-Range Imaging for Cloud Segmentation" Soumyabrata et al., present HDR method for separation of cloud and clear-sky pixels detected by whole-sky cameras at different exposure time. This method is alternative for whole sky camera which used the moving arm to block the direct solar radiation and part of solar aureole. Although authors cited the Long et al., 2006, there is no discussion about this method in the manuscript. Such technique is used in the whole sky cameras developed by Yankee company ( http://www.yesinc.com/products/data/tsi880/tsi-880ds.pdf ) to reduce the direct solar radiation measured by the CCD sensors. Presented method in this manuscript can be useful for cloud and aerosol community. For example measurements of aerosol optical depth required clear sky condition close to sun. Detection of thick clouds is

not a problem but cloud screening of the thin cirrus or cirrus subvisiual is not trivial. Therefore discussed of different sky condition (thick and thin clouds) is very important for this kind of the algorithm. I would like to see how the algorithm works with thin cirrus close to sun aureole. In my opinion, the manuscript can be publishing in AMT after minor revision.

Specific comments: 1. Some references to the first part of the introduction in needed (e.g. IPCC, 2013, Stephens et al., 2012 NATURE GEOSCIENCE | VOL 5 | OCTOBER 2012 | ) 2. Line 13: instead of "meteorological centers" -> WMO (World Meteorological Organization) stations 3. Line 20: Could you add reference to sky camera at UCSD in San Diego?
* * *

---

## Referee Comment (RC2) · Anonymous Referee #3 · 3 Jan 2018

In the manuscript a simple yet efficient method of improving a dynamic range of sky camera is described and discussed. The authors use standard bracketing to capture three consequent images at three various exposures and proces them by contrast-limited adaptive histogram equalization algorithm and further fuzzy logic and probabilistic image segmentation, improving quality of the final image. In particular the method substantially reduces number of saturated pixels and benchmark tests show its advantage over other post-processing methods described in the literature. The text is clearly written and contains all the necessary information, however in the presentation there are some elements which should be improved. Thus, the paper can be accepted to AMT after minor revisions.

Specific comments.

[Figure]

1) Figures shall be page wide in the final version of the manuscript.

2) Figure 5: any ideas why such a range of segmentation errors in various colour channels? A short explanation is necessary, the reviewer has some ideas why C15 is the best choice, but this should be explained in more detail.

3) Figure 6. Any ideas why there are dips and tops on presented curves? Explain, please.

---

## Author Comment (AC1) · 14 Feb 2018

**Response to Reviewers' Comments**
**amt-2017-152 High-Dynamic-Range Imaging for Cloud Segmentation**

Soumyabrata Dev, Florian M. Savoy, Yee Hui Lee, Stefan Winkler

February 14, 2018

We would like to thank the Associate Editor Prof. Szymon Malinowski and the anonymous referees for your valuable comments and suggestions.

Based on your inputs, we have thoroughly revised the manuscript. All the comments and suggestions have been addressed. Responses to the individual comments can be found below. Unless otherwise specified, the references, equations, figures and tables cited in the answers are numbered as per the revised manuscript.

We are releasing the first HDR dataset of sky/cloud images, along with its manually annotated ground-truth images.

The source code of all simulations in this paper is also released, and is now available online at `https://github.com/Soumyabrata/HDR-cloud-segmentation`.

```
>> ANONYMOUS REFEREE #1<<
In the manuscript High-Dynamic-Range Imaging for Cloud Segmentation Soumyabrata et al.,
    present HDR method for separation of cloud and clear-sky pixels detected by whole-sky
     cameras at different exposure time. This method is alternative for whole sky camera
     which used the moving arm to block the direct solar radiation and part of solar
     aureole.
```

Thank you for your positive feedback on the manuscript.

```
Although authors cited the Long et al., 2006, there is no discussion about this method
    in the manuscript. Such technique is used in the whole sky cameras developed by
    Yankee company (http://www.yesinc.com/products/data/tsi880/tsi-880ds.pdf) to reduce
    the direct solar radiation measured by the CCD sensors.
```

Thank you for the suggestion. We have included more discussion in the manuscript, on the commercial sky camera, manufactured by Yankee Environmental Systems.

We have included this discussion in Section 1 of the manuscript.

```
Discussed example of the sky condition is very simple (cloud with high optical depth,
    probably Cu or Sc) for this kind of the algorithms. I would like to see how the
    algorithm works with thin cirrus close to sun aureole. Detection of thick clouds is
    not a problem. Presented method in this manuscript can be useful for cloud and
    aerosol community. For example measurements of aerosol optical depth required clear
    sky condition close to sun. Detection of thick clouds is not a problem but cloud
    screening of the thin cirrus or cirrus subvisiual is not trivial. Therefore discussed
     of different sky condition (thick and thin clouds) is very important for this kind
    of the algorithm. I would like to see how the algorithm works with thin cirrus close
    to sun aureole.
```

Thank you for the feedback on the cloud detection performance of the proposed algorithm around the sun aureole. In this manuscript, we are essentially dealing with a binary classification problem, where pixels are classified as either *sky* or *cloud* pixels. In the proposed method, the detection of

the clouds is dependent on the seeding parameter $\alpha$. In order to illustrate this fact, we generate the Receiver Operating Characteristics (ROC) curve for varying values of the seeding parameter. Figure 1 shows the ROC curves for varying values of $\alpha$, in the range $[0, 1]$.

[Figure]

Figure 1: Impact of the seeding parameter on Receiver Operating Characteristics (ROC) Curve; the *diamond* data points indicate the intermediate ROC points, for seeding parameter in intervals of 0.1 in the range [0,1].

This sensitivity of the seeding parameter can also help us in understanding the different type of clouds around the sun aureole. In this manuscript, we deal only with the detection of thick clouds in the circumsolar region. Primarily, this is because, classifying the type of clouds into *thick* or *thin* clouds is an extremely difficult task in the region around the sun. Most of circumsolar region pixels are saturated, owing to its exposure to sun. However, the seeding parameter $\alpha$ has a control on the type of detected clouds. We can detect thin clouds by tuning the seeding parameter, to favor high cloud detection (i.e. more tendency to classify a pixel as cloud).

Our proposed algorithm, HDRCloudSeg works best around the circumsolar region, as compared to the state-of-the-art algorithms. To illustrate this fact, we provide a visual interpretation of the result in Fig. 2.

[Figure]

Figure 2: Illustration of HDRCloudSeg performance near the sun aureole. (a, b, c) Cropped low-, mid- and high- exposure Low-Dynamic-Range (LDR) images in the circumsolar region; (d, e, f) Corresponding low-, mid- and high- exposure LDR images, with saturated pixels marked in pink; (g) Tonemapped High-Dynamic-Range (HDR) image; (h) Seeded image, where definite cloud pixels are marked in *green*, and definite sky pixels are marked in *red*; (i) Binary output using HDRCloudSeg.

We show a $300 \times 300$ cropped image around the sun. The cropped low-, mid-, and high- exposure LDR images are shown in Fig. 2(a, b, c). We show the corresponding saturated pixels in these different LDR images in Fig. 2(d, e, f). It is interesting to observe that our proposed HDRCloudSeg approach starts seeding the HDR tonemapped image with a high level of accuracy. The pixels around the sun aureole remain unseeded, as they are most difficult to classify. Also, the boundary pixels between sky and cloud also remain unseeded. The graph cut module in HDRCloudSeg, subsequently classifies the unseeded pixels into sky and cloud category accordingly. Such behavior of our algorithm in the circumsolar region demonstrates the superiority of our approach, near the sun aureole region.

In this manuscript, we have not considered the detection of a thin clouds (viz. multi-class classification problem). This is a non-trivial task, and furthermore, the cloud experts from Singapore Meteorological Center have assisted us in generating the binary ground-truth maps of the HDR image sets. In the future work, we will generate a ternary ground-truth map (consisting of *clear sky*, *thick cloud* and *thin cloud*), in consultation with cloud experts. This will help us in a quantitative evaluation of thin cloud detection around the sun aureole.

We have indicated these changes in Section 4.2.2 and Section 5 of the manuscript.

`In my opinion, the manuscript can be publishing in AMT after minor revision.`

Thank you for the positive feedback on the manuscript.

Specific comments: 1. Some references to the first part of the introduction in needed (e.
    g. IPCC, 2013, Stephens et al., 2012 NATURE GEOSCIENCE | VOL 5 | OCTOBER 2012 | )

Thank you for the suggestions. In order to make the introduction of the manuscript more inclusive on other atmospheric studies, we have cited the following publications.

- Climate Change 2013: The Physical Science Basis, Intergovernmental Panel on Climate Change (IPCC).

- Stephens, G.L., Li, J., Wild, M., Clayson, C.A., Loeb, N., Kato, S., Lecuyer, T., Stackhouse Jr, P.W., Lebsock, M. and Andrews, T., 2012. An update on Earth's energy balance in light of the latest global observations. Nature Geoscience, 5(10), p.691.

We have included a concise discussion of these publications in Section 1 of the manuscript.

2. Line 13: instead of meteorological centers -> WMO (World Meteorological Organization)
    stations

Thank you for the suggestion. We have edited this phrase 'meteorological centers' in Section 1 of the manuscript.

3. Line 20: Could you add reference to sky camera at UCSD in San Diego?

Thank you for the suggestion. We have added a reference to the sky camera used at University of California, San Diego (UCSD).

We have edited Section 1 of the manuscript to indicate this update.

---

## Author Comment (AC2) · 14 Feb 2018

**Response to Reviewers' Comments**
**amt-2017-152 High-Dynamic-Range Imaging for Cloud Segmentation**

Soumyabrata Dev, Florian M. Savoy, Yee Hui Lee, Stefan Winkler

February 14, 2018

We would like to thank the Associate Editor Prof. Szymon Malinowski and the anonymous referees for your valuable comments and suggestions.

Based on your inputs, we have thoroughly revised the manuscript. All the comments and suggestions have been addressed. Responses to the individual comments can be found below. Unless otherwise specified, the references, equations, figures and tables cited in the answers are numbered as per the revised manuscript.

We are releasing the first HDR dataset of sky/cloud images, along with its manually annotated ground-truth images.

The source code of all simulations in this paper is also released, and is now available online at `https://github.com/Soumyabrata/HDR-cloud-segmentation`.

```
>> ANONYMOUS REFEREE #3<<
In the manuscript a simple yet efficient method of improving a dynamic range of sky
   camera is described and discussed. The authors use standard bracketing to capture
   three consequent images at three various exposures and proces them by contrast-
   limited adaptive histogram equalization algorithm and further fuzzy logic and
   probabilistic image segmentation, improving quality of the final image. In particular
    the method substantially reduces number of saturated pixels and benchmark tests show
    its advantage over other post-processing methods described in the literature. The
   text is clearly written and contains all the necessary information, however in the
   presentation there are some elements which should be improved. Thus, the paper can be
    accepted to AMT after minor revisions.
```

Thank you for your positive feedback on the manuscript.

```
Specific comments.
1) Figures shall be page wide in the final version of the manuscript.
```

Thank you for the suggestion. In this revised version, we ensured that all figures contain the entire width of the page.

```
2) Figure 5: any ideas why such a range of segmentation errors in various colour
   channels? A short explanation is necessary, the reviewer has some ideas why C15 is
   the best choice, but this should be explained in more detail.
```

The existing approaches in the literature, uses a combination of *red* and *blue* color channels for cloud segmentation. It is due to a physical phenomenon called Rayleigh scattering. The small particles in the atmosphere scatter light at varying degree. The component of white light having the least wavelength (blue component) gets scattered the most. This renders a bluish color to the sky. The $c_{15}$ color channel is the normalized ratio of *red* and *blue* color channels; and is the most *discriminatory* feature for cloud detection. This is a sensible choice, because the sky is predominantly blue in color.

In an earlier publication [S. Dev, Y. H. Lee, S. Winkler, Systematic Study of Color Spaces and Components for the segmentation of sky/cloud images, *Proc. IEEE International Conference*

*on Image Processing (ICIP)*, 2014], we have provided a systematic analysis of the various color channels for cloud detection. Using a set of statistical tools, and a data-centric approach, we concluded in our earlier publication, that $c_{15}$ is a good color channel for conventional 8-bit low-dynamic-range images too. In this manuscript, we observe this behavior too, and conclude that $c_{15}$ is the best color channel for HDR sky/cloud images too.

We have added a discussion on the same in Section 4.2.1 of the revised manuscript.

**3) Figure 6. Any ideas why there are dips and tops on presented curves? Explain, please.**

Thank you for the feedback. Although the general trend of the curve is consistent w.r.t. the increasing seeding level, there are a few deviation points along the curve. The minor *peaks* and *troughs* in Figure 6 are because of the sensitivity of the considered seeding level. This causes error in the seeding accuracy, that subsequently impacts the final evaluation metric of cloud detection.

We have edited Section 4.2.2 of the revised manuscript, to indicate this change.